# Comparing Accuracy of Implant Installation with a Navigation System (NS), a Laboratory Guide (LG), NS with LG, and Freehand Drilling

**DOI:** 10.3390/ijerph17062107

**Published:** 2020-03-22

**Authors:** Ting-Mao Sun, Huey-Er Lee, Ting-Hsun Lan

**Affiliations:** 1School of Dentistry, College of Dental Medicine, Kaohsiung Medical University, Kaohsiung 80756, Taiwan; dms.timax@gmail.com (T.-M.S.); iandyou_snow@hotmail.com (H.-E.L.); 2Division of Family Dentistry, Department of Dentistry, Kaohsiung Medical University Hospital, Kaohsiung 80708, Taiwan; 3Division of Prosthodontics, Department of Dentistry, Kaohsiung Medical University Hospital, Kaohsiung 80708, Taiwan

**Keywords:** accuracy, computer-guided, dental navigation system, implant surgery

## Abstract

The aim of this study was to compare the accuracy of implant placement by using the conventional freehand method, the surgical guide alone, the dental navigation system alone, and the dental navigation system with a surgical guide. The participants were aged 20 years or older and were requiring dental implant surgery according to an assessment made by a dentist between July 2014 and December 2017. A total of 128 dental implants were inserted, 32 dental implants in each group, and participants with similar or identical age (i.e., 20–50 years or 50 years or above) and missing tooth locations were paired for comparison. Accuracy was measured by overlaying the real position in the postoperative Cone Beam Computerized Tomography (CBCT) on the virtual presurgical placement of the implant in a CBCT image. Using the dental navigation system with a surgical guide could help dentists to position implants more accurately. Total, longitudinal, and angular error deviation were significantly different (*p* < 0.0001). The same level of accuracy could be obtained for the different jaws and tooth positions. The one-way analysis of variance (ANOVA) showed that the total, longitudinal, and angular errors differed significantly (*p* < 0.0001). A comparison of the four dental implant surgical methods indicated that the combination of a dental implant navigation system and a surgical guide kit achieved the highest accuracy in terms of the different tooth positions and jaws.

## 1. Introduction

Advances in artificial dental implant technology have contributed to high dental implant success and survival rates, making dental implants a viable option in modern dental treatment. When patients experience problems such as missing teeth, loss of occlusion, difficulty with chewing, and pronunciation, and unfavorable oral appearance, their preference is often to receive an artificial dental implant [1]. However, many risk factors of dental implant surgery result in irreversible complications, including in anatomical sites (e.g., neural tube damage and alveolar bone perforation) and transplantation sites (e.g., implants ending up at an angle or too close to adjacent teeth) [2]. To accurately place an implant at a transplantation site, dentists could use medical imaging to make related decisions. The evolution of three-dimensional (3D) imaging now enables doctors to visualize the structure of the dissection site and accurately detect related pathological symptoms. Therefore, 3D imaging has been widely used in preoperative planning and during dental implant surgery [3,4].

Digital technology is changing with each passing day. In 1995, Fortin et al. [5] proposed computer-assisted implant surgery (CAIS), which involves placing implants in a relative position through preoperative planning. Computer-assisted surgery (CAS) for dental implant placements includes static and dynamic systems. Static systems use computerized tomography–generated computer-aided design/computer-aided manufacturing stents, with sleeves and a surgical system that uses coordinated instrumentation to place implants by using a guide stent [6]. Dynamic CAS systems track the patient and surgical instruments and present real-time positional and guidance feedback on a computer display [7]. Dental navigation systems, which have been developed further in conjunction with surgery equipment, medical imaging, optical positioning devices, and dental implant preoperative planning software, can now offer optical positioning technology and guide its user to the planned location through a clinically applicable interface [4]. The optical systems use passive or active tracking arrays. Passive systems use tracking arrays that reflect light emitted from a light source back to stereo cameras [8]. Active system arrays emit light that is then tracked by the stereo cameras [9]. The drill and patient-mounted array must be within the line of sight of the overhead stereo cameras to be accurately tracked on the monitor [10]. Wittwer et al. [11] and Poeschl et al. [12] indicated that using jaw-mounted tracking systems is effective for considering patient movements during surgery. Consequently, Block et al. [13] analyzed the precision and accuracy of dental implant navigation systems, and demonstrated that these systems are more precise and accurate than is traditional freehand dental implant surgery. Nonetheless, users were advised to gain experience of these systems and reach the learning curve, after which their dental implant surgery precision and accuracy can increase [13,14,15].

For traditional implant surgery, dentists perform numerous surgical procedures to accumulate experience from patients with different oral cavity conditions and learn to use dental implant drilling with their free hands. Relying only on experience in general surgery can result in various medical errors [16,17]. Moreover, the disadvantages of using guide stents alone in a dental implant surgery include uncertainty of depth and edentulous conditions [18,19]. Pernar et al. [15] also reported that the presence of bone spine causes drills to move up and down unsteadily when using dental implant navigation systems alone and thus results in errors. However, Chen et al. [20] showed that the dental implant navigation system enables dentists to identify insertion depths and the location of the guide stents with sleeves, reducing the incidence of dental implant devices moving up and down. 

In a systematic review, Widmann et al. [21] reported that dental navigation systems have the following advantages: (1) Because only scanning, designing, and preoperative planning are required, surgery duration can be reduced. (2) During surgery, the surgical plan and the size of the implant can be changed according to the actual clinical situation. (3) The surgeon can improve their accuracy and reduce surgical damage by being able to react instantly to the positional distance between the drill and the anatomical structure [22]. However, the disadvantages of dental navigation systems include the following: (1) Expensive equipment is required. (2) Large clinical surgical space is required. (3) The tracking system affects the surgical field of vision and the operational feeling on hands [11]. 

In addition, Schneider and Tahmaseb et al. [6,23] have conducted systematic analyses on the accuracy of static surgical guides in 2009 and 2014, respectively; the authors indicated the advantages and disadvantages of using surgical guides. The advantages of surgical guides were as follows: (1) shortened surgical duration, (2) no flaps or small wounds created, and (3) ability to place dental implants in the correct positions. However, the disadvantages of surgical guides were as follows: (1) relatively high cost, (2) possible template sliding due to use of different supporting methods, and (3) errors caused by the gap between the guide and drill bit (the incidence of which may increase further as the drilling distance increases).

Implant surgery has long involved dental implant navigation systems and surgical guide stents [6,7]. The hypothesis of this study is that dentists with relatively extensive dental surgery experience do not necessarily have higher accuracy in performing dental implant surgery by using the relevant auxiliary systems. The purpose of this study recruited the patients who required dental implants (according to clinical assessments) to compare the differences in accuracy between four dental implant surgery techniques, namely a dental navigation system with a surgical guide, a dental navigation system, a surgical guide, and freehand.

## 2. Materials and Methods

This study has been approved by the Human Research Ethics Committee (IRB code: KMUHIRB-2013-08-02(I) and IRB code: 20160201A). Informed consent was acquired by all patients. The study followed the CONSORT statements for reporting clinical trials.

### 2.1. Main Operator

Over the past two decades, the recruited dentist has performed dental implant surgeries using the traditional freehand technique. After the dentist had been introduced to a dental implant navigation system, he completed five preoperative training tests (including 150 maxillary and mandible drilling tests) to verify whether he had reached the required levels of precision and accuracy using the system. Since 2014 (as approved by the institutional review board), the dentist performed four dental implant surgery techniques (dental navigation system with surgical guide, dental navigation system, surgical guide and freehand) in dental clinic department.

### 2.2. Participants Recruitment

In this study, the recruitment criteria were age of 20 years or older and requirement of dental implant surgery according to an assessment made by a dentist. The exclusion criteria were that patients with cardiovascular diseases, those with severe alveolar bone defects, those who had received head or neck radiotherapy, those with severe diabetes, and those who could not sign the consent form because they did not understand its content. All participants agreed to participate in the study and signed a consent form to protect their safety and rights. Participants who refused to sign the consent form were excluded from the study and underwent surgery using the traditional dental implant method. The study participants were divided into experimental and control groups, and four surgical methods were employed. The experimental groups have three subgroups in which a dental implant navigation system combined with surgery guide stent set were used, the dental implant navigation system alone was used, and the surgery guide stent set alone was used. The control group underwent surgery using the traditional dental implant free hand method. A total of 32 dental implants were inserted in each group, and participants with similar or identical age (i.e., 20–50 or >50 years), maxillary and mandible positions, and missing tooth locations were paired for comparison. A total of 128 dental implants were inserted.

### 2.3. Power Calculation

The minimum required sample size of implants according to total, longitudinal, and angular deviation, respectively, was separately calculated using a statistical software (Stata Statistical Software, Stata Corp LP, College Station, TX, USA). for One-Way ANOVA F test with 80% of study power and significant level (α) of 0.05.

### 2.4. Four Dental Implant Surgery Approach

#### 2.4.1. Navigation System

The dental navigation system (AQNavi, TITC Ltd, Kaohsiung, Taiwan) is an electro-optical device, which was specifically designed to greatly improve the surgical implantation procedure by providing the surgeon an accurate guidance to the location of the surgical tools prior to, and more importantly, during the surgical operation. This system is a computer-based assistance system for dental implant surgeon. It supports surgeons during planning and execution phase of dental implant procedures by providing highly precise guiding assistance during the implantation process within the human jaw and providing visual guidance for position and orienting of a dental drill relative to a planned implant position.

#### 2.4.2. Computer-Guide 

The surgical guide uses a photo-curing resin that has been tested by biocompatibility to make a positioning collar with a medical grade titanium alloy. Before the production, the patient should first take a full-computer computed tomography (CBCT) to obtain images of the teeth and bones, through a tooth model or a 3D file of the teeth obtained by intraoral scanning, which can be combined to simulate the position of the teeth and bones. The DICOM files from the CBCT scan were converted to preoperative planning software (SmilePlan, TITC Ltd., Kaohsiung, Taiwan). Through the dental implant planning software, the most appropriate implant size and implant position, angle and depth, and then design a suitable surgical guide were found, which could be directly processed by CAD/CAM technology.

#### 2.4.3. Freehand 

The implant was placed depending on the perspective of the surgeon within the same image analysis and treatment plan decision. 

### 2.5. Postsurgical Care

All patients were given antibiotics (1 g of amoxicillin, two times a day, for 5 days) and analgesic drugs (500 mg of mefenamic acid, three times a day, for 5 days). Postimplant placement CBCT scans with the aforementioned settings were taken 1 week after surgery. For the dynamic CAIS group, the same registration stent and fiducial-containing devices were inserted during the scan.

### 2.6. Operational Processes for Four Approaches 

Figure 1 showed the workflow of participant recuitment and operational processes for four approaches.

#### 2.6.1. First Step: Prosthodontic and Computed Tomography (CT) Stage

Dental impressions were made for all participants and were used to develop CT stents and guided stents for photo shooting. After repeated tests, the patients tried on the CT stents and guided stents and adjusted them for correct fit. Subsequently, the participants underwent dental CT scanning (voxel size = 0.15 × 0.15 × 0.15 mm^3^, AZ3000CT, Asahi, Japan). Cone beam CT was performed to obtain DICOM files with standard formatting. 

#### 2.6.2. Second Step: Preoperative Planning

The CT scan images were inputted to a dental implant preoperative planning software called SmilePlan (developed by TITC Ltd., Kaohsiung, Taiwan). This software determined the optimal dissection site structure of the missing teeth area as well as the relative locations of nearby teeth. The appropriate dental implants were then planned for the surgery locations.

#### 2.6.3. Third Step: Preoperative Calibration for Navigation System and Navi with Guide

The handpiece and patient tracking module were calibrated by an engineer prior to each surgery. Each drill length was calibrated because it was used by the surgeon in the normal sequence of implant site preparation. System checks were performed to ensure the accuracy of tracking, and the implant was guided into its final position using the navigation screen. A position indicator was displayed as a real-time traceable red cross; the center of the screen designated the planned position. The coordinates of the system were designed to accommodate both the buccal-lingual and mesial-distal directions, making it easy for all dentists to accurately read and interpret the location data. The angle indicator used a real-time traceable ball to indicate the current angular deviation, whereas the center of the screen indicated the planned angle. Window coordinates were also designed to accommodate both the buccal-lingual and mesial-distal directions. The depth indicator used a bar to indicate the distance to the correct position of the apex. A pane in the upper right corner of the depth indicator showed whether the depth of the drill of the handpiece exceeded the correct depth (Figure 2).

#### 2.6.4. Fourth Step: Postoperative Planning and Preoperative and Postoperative Integration

After surgery, the patient was required to wear the CT stents for new CT scanning. The new CT images were then inputted to commercial software (Solidworks 2013 for Windows, Dassault Systems SolidWorks Corp., Waltham, MA, USA) to generate preoperative and postoperative spatial overlays. A built-in measurement function was used to measure the spatial and angular deviation before and after the surgery. Because the ceramic beads of the CT stents did not scatter when receiving CT scans, SmilePlan (developed by TITC Ltd., Kaohsiung, Taiwan) as used to identify the balls’ locations and record them in CT coordinates. The coordinates were converted into a 3D coordinate system using Solidworks.

The preoperative and postoperative spatial coordinate files were input as combined Solidworks files. During positioning, the CT stents of the two coordinate files were combined by their points (i.e., the oral positioning point of SmilePlan (built-in point no. 3), line (i.e., the second longest side of the triangle formed by the three outermost points of the oral positioning point), and plane (i.e., the triangle formed by the three outermost points of the oral positioning point)).

New Solidworks functions were used to construct coordinate axes on implant axes. Measurement functions were used to measure the lower endpoints of the implants inserted, obtaining values dx, dy, and dz. By taking the square root of dx and dy, the postoperative distance deviation was derived. Similarly, postoperative depth deviation was obtained using dz.

The SmilePlan oral positioning point (built-in point no. 3) was changed to the upper endpoints for measuring the deviation of the two implants. After the two endpoints had been combined, the upper endpoints, lower endpoints of the implants planned before the operation, and upper endpoints of the inserted implants were used to construct a plane. This plane was used to plot the axes planned before the operation and the postoperative implant insertion axes. Subsequently, Solidworks was used to measure the postoperative angle deviation.

### 2.7. Surgery Process

The surgeon clearly realized the limitation of anatomy and risk of all the cases. All participants were flap reflection in the implant position under anesthesia, and the surgeon could directly look at the alveolar bone in the operation area to understand the bone condition. Use dental implant drills to drill holes in the alveolar bone and select the appropriate implant fixture (MaxFit, TITC Ltd., Kaohsiung, Taiwan) into the bone cavity until the implant is completely immersed in the alveolar bone, and then suture. After 7 to 14 days, the patients needed to return the dental clinic to remove the suture and take care about the wound.

### 2.8. Deviation Definition

After collection of the data and determination of the coincidence method, the relative error value between preoperative planning (Master) and the actual implant (Slave) was calculated. The following is the related explanation for the total error, longitudinal error, and angular error (Figure 3):Total Error: Point A1 was set as the origin, and vector A ($\vec{A}$) was the Z-axis in space. The Pythagorean theorem was applied to obtain the 2-dimensional error between vector B ($\vec{B}$) and vector A.Longitudinal Error: Point A1 was set as the origin, and vector A ($\vec{A}$) was the Z-axis in space. The coordinate difference of the 2 points along the Z-axis was the longitudinal error.Angular Error (ø): The product of spatial vectors used to obtain the angle between vector A ($\vec{A}$) and vector B ($\vec{B}$) was the angular error.

### 2.9. Statistical Methods

The mean and standard deviation among the 3 errors (total, longitudinal, and angular errors) were calculated through one-way ANOVA, followed by the Tukey–Kramer multiple comparison test. The data gained from this study were analyzed by using the t-test between all deviation errors and different jaws and tooth positions. Analyses were performed using the JMP statistical software, version 14.0.0 (JMP, SAS Institute, Inc., Cary, NC, USA). Power calculated using a statistical software (Stata Statistical Software, Stata Corp LP, College Station, TX, USA).

## 3. Results

Table 1 shows the results of these four surgical tests in the deviation of the total, longitudinal, and angular errors. The comparison among the tests performed by different surgical approaches also demonstrated a significant difference. (*p* < 0.0001 for the total error, *p* < 0.0001 for the longitudinal error, and *p* < 0.0001 for the angular error). In terms of the time of preparation for surgery, it can be seen that the time for preparation of the freehand surgery is shorter, compared with the dental navigation system with the surgical guide. According to the one-way ANOVA and Tukey–Krammer HSD tests, the differences among all surgical approach showed significant differences, and total error, longitudinal error, and angular errors were significant.

Table 2 shows the results of the deviation of the total, longitudinal, and angular errors between the experimental and control group in different jaw and positions. According to the two-sample *t*-tests, the differences among all surgical approaches showed were significant, and the total error, longitudinal error, and angular error were also significant. In addition, the angular error in the experimental group was not significant between the different jaws.

Table 3 shows the deviation of the total, longitudinal, and angular errors in four surgical approaches by different jaws and dental positions. The one-way ANOVA and Tukey–Kramer HSD tests revealed that the total, longitudinal, and angular errors differed significantly among different jaws and tooth positions. Freehand dental implant surgery has poorer accuracy than other surgical methods. The dental implant navigation system and surgical guide kit achieved the highest accuracy in terms of the different tooth positions and jaws.

## 4. Discussion

This study demonstrated that both dynamic and passive dental implant guidance systems can aid dentists in increasing dental implantation accuracy in clinical practice, with differences in total, longitudinal, and angular errors among the four dental implant surgery techniques. The current findings rejected the study hypothesis and the accuracy of surgery using the combination of the dental navigation system and surgical guide kit, using the dental navigation system alone, or using the surgical guide alone, was greater than that of using conventional freehand surgery alone.

Block et al. [24] concluded that the results obtained using dynamic guidance systems are more accurate than those obtained using passive guidance systems—supporting the results of the current study. Somogyi-Ganss et al. [25] also demonstrated that the results obtained using a novel prototype of a dynamic CAS system were more accurate than those obtained using an acrylic surgical guide. Chen et al. [20] compared the accuracy achieved by using the navigation system, a surgical guide, and freehand surgery in vitro; the accuracy achieved when using the navigation system was superior to that achieved when using a surgical guide and far superior to that achieved through conventional dental implantation. This finding is in agreement with those of the current study. Moreover, the present study determined that a combination of the navigation system and a surgical guide yielded the optimal accuracy among the different surgical methods. Tallarico et al. [26] showed that dynamic CAS systems allow more accurate implant placement compared with the conventional freehand method, regardless of the surgeon’s experience. The present study demonstrated the accuracy of dental navigation systems and the conventional freehand method. The total longitudinal and angular error demonstrated that navigation systems are more accurate than freehand insertion. Emery et al. [27] proposed that after guided osteotomy, guided implant placements had accuracy superior to that of freehand placements. In the present study, the average deviation provided information on the implantation accuracy of the surgical guide and of using freehand; the corresponding averages for the surgical guide demonstrated higher accuracy than that for freehand implantation.

This study observed significant differences between the maxillary and mandibular outcomes obtained using the four dental implant surgery approaches. The accuracy of the surgery to the mandible was superior to those of the maxillary. Casap et al. [28] demonstrated that computerized navigation systems enable increased accuracy for lower-jaw surgery. In an in vivo study, Vasak et al. [29] found lower deviations for surgery in the mandible. Implants in the frontal region of the mandible had the highest accuracy, followed by those of the posterior region of the mandible, whereas those of the posterior region of the maxilla had the lowest accuracy. Fang et al. [30] noted that because people can only open their mouths to a certain extent and have buccal soft tissue, surgery using a dental implant device can be performed in limited spaces, resulting in results of posterior teeth surgery being inferior to those of anterior teeth surgery. Tallarico et al. [26] observed that anterior implants had higher accuracy than posterior ones based on all deviation errors.

Considering all participants in separate experimental and control groups, this study reported that the total, longitudinal, and angular errors in the experimental group were superior to those in the control group. After using the dynamic and static systems, all groups, including the dentist with extensive surgery experience, demonstrated increased accuracy. Before using these systems, dentists faced a learning curve in their knowledge of the systems. After reaching a learning curve plateau, the dentists demonstrated the accuracy achievable by using these systems. Sun et al. [31] indicated that dentists must have received a certain amount of in vitro training before they could clinically use the systems with precision and accuracy and prevent human factors from compromising system performance. 

Human factors can affect the performance of the computer-assisted implant systems, particularly in dental navigation systems. For successful transfer of preoperative planning to the clinical operation through the dental implant navigation system, a surgeon is required to have eye–hand coordination, such that the data of the immediate monitoring of the navigation system and handling of the drilling process during surgery can be determined [25]. Rungcharassaeng et al. [32] reported the vertical deviations caused by inexperienced dentists were more than twice than those caused by experienced dentists. Through a similar test, Cushen et al. [33] demonstrated that the vertical and angular deviations of inexperienced dentists were worse than those of experienced dentists. Cassetta et al. [34] demonstrated no significant differences between experienced and inexperienced dentists. To reduce the influence of human factors in this study, the main operator conducted in vitro tests to reach the learning curve and then joined the clinical experiment. These test data are the closest to the original system error.

The present study had three limitations: (1) only one dentist was investigated in the comparison of accuracy achieved using different surgical methods; (2) more cases should have been included to verify the experimental data in the discussion of the dental tooth positions; and (3) the experimental results would have been more reliable if a triple trial had been conducted. In future research studies, more dentists and patients with missing teeth can be recruited to further clarify the clinical accuracy of various surgical methods, thereby assisting dentists in providing more comprehensive medical care programs and enhancing care quality.

## 5. Conclusions

This study mainly wants to understand the accuracy of the same dentist under different auxiliary systems. The results were analyzed by statistical power, one-way ANOVA, and ANOVA Tukey–Kramer HSD to ensure the representativeness and consistency. The conclusions show the results of this study on the accuracy of the dental navigation system alone is superior to surgical stent alone and freehand surgery alone. A comparison of the four dental implant surgical methods indicated that the combination of the dental implant navigation system and surgical guide kit achieved the highest accuracy in terms of the tooth positions and upper and lower jaws.

## Figures and Tables

**Figure 1 ijerph-17-02107-f001:**
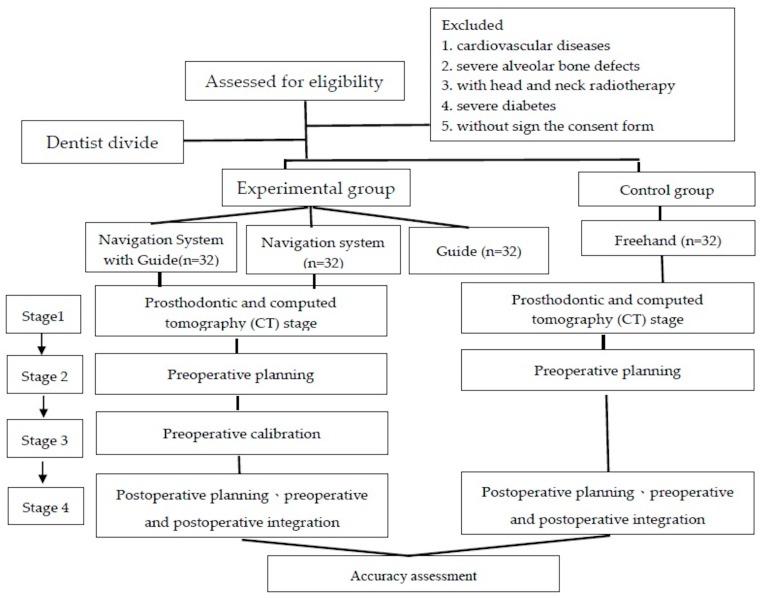
The workflow of participant recruitment and operational processes for four approaches.

**Figure 2 ijerph-17-02107-f002:**
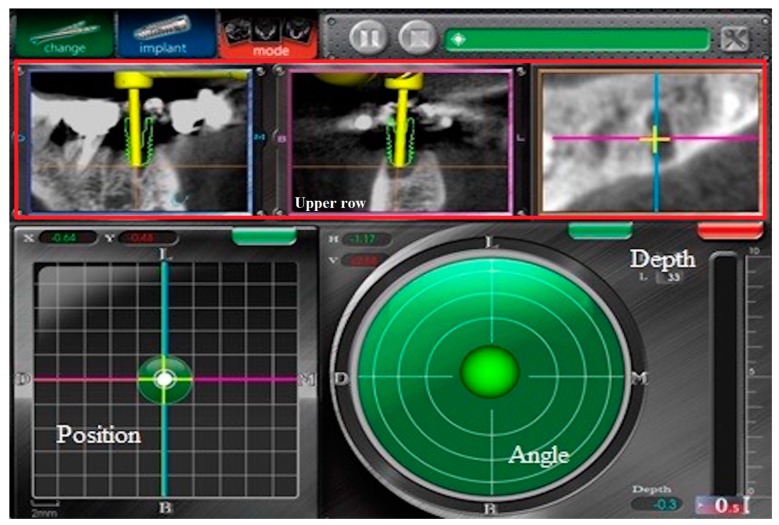
The monitor shows the depth, position, and angle of dental implant. The upper row shows the real-time labial view, sagittal view, and occlusal view (upper row from left to right). This figure was authorized by Taiwan Implant Technology Company.

**Figure 3 ijerph-17-02107-f003:**
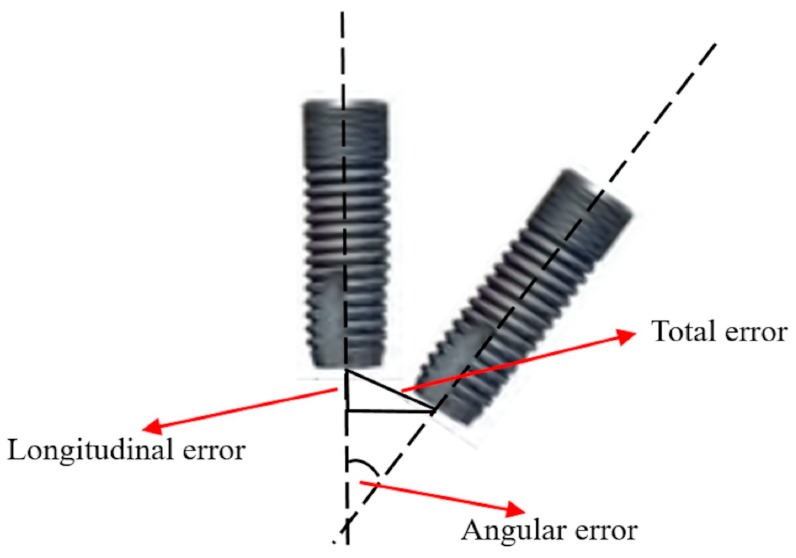
Total error (mm), longitudinal error (mm), and angular error (degree) measurement.

**Table 1 ijerph-17-02107-t001:** The deviation of the total, longitudinal, and angular errors in the different surgical approach.

Deviation	Navigation and Surgical Stent		Navigation		Surgical Stent		Freehand		*p*-Value *	Multiple Comparison
Error	Mean ± SD	95% CI	Mean ± SD	95% CI	Mean ± SD	95% CI	Mean ± SD	95% CI		
Total (mm)	0.98 ± 0.19	0.91–1.05	1.25 ± 0.09	1.22–1.28	1.49 ± 0.08	1.47–1.53	1.89 ± 0.09	1.86–1.93	<0.0001 #	NS > N > S > F
Longitudinal (mm)	0.52 ± 0.20	0.44–0.59	0.73 ± 0.13	0.68–0.77	1.00 ± 0.15	0.96–1.06	1.42 ± 0.25	1.51–1.33	<0.0001 #	NS > N > S > F
Angular (degree)	2.20 ± 0.38	2.06–2.34	3.24 ± 0.36	3.11–3.38	4.54 ± 0.29	4.43–4.65	6.12 ± 0.12	6.08–6.17	<0.0001 #	NS > N > S > F
Prepare Time	30 min		20 min		15 min		10 min			

CI: Confidence Level; *: Using one-way ANOVA; #: Tukey–Kramer HSD; NS: Navigation and Surgical stent; N: Navigation; S: Surgical stent; F: Freehand.

**Table 2 ijerph-17-02107-t002:** The deviation of the total, longitudinal, and angular errors in experimental and control groups by different jaw and dental position.

	Total Error (mm)			Longitudinal Error (mm)			Angular Error (degree)		
	Mean ± SD	95% CI	*p*-Value *	Mean ± SD	95% CI	*p*-Value *	Mean ± SD	95% CI	*p*-Value *
Experimental group									
Maxillary	1.15 ± 0.26	1.08–1.23	0.0003 *	0.63 ± 0.24	0.56–0.69	<0.0001 *	3.07 ± 0.99	2.78–3.36	0.0123 *
Mandible	1.33 ± 0.20	1.27–1.39		0.88 ± 0.22	0.81–0.94		3.59 ± 0.99	3.29–3.88	
Anterior	1.19 ± 0.26	1.11–1.26	0.0377 *	0.69 ± 0.26	0.61–0.76	0.0150 *	3.19 ± 1.02	2.89–3.49	0.2029
Posterior	1.29 ± 0.22	1.23–1.36		0.81 ± 0.25	0.74–0.89		3.46 ± 1.01	3.17–3.76	
Control group									
Maxillary	1.82 ± 0.04	1.79–1.84	<0.0001 *	1.21 ± 0.09	1.16–1.25	<0.0001 *	6.02 ± 0.04	6.01–6.05	<0.0001 *
Mandible	1.97 ± 0.07	1.93–2.01		1.63 ± 0.17	1.53–1.71		6.22 ± 0.09	6.17–6.27	
Anterior	1.85 ± 0.07	1.81–1.88	0.0050 *	1.31 ± 0.19	1.21–1.41	0.0170 *	6.07 ± 0.09	6.03–6.12	0.0130 *
Posterior	1.94 ± 0.09	1.88–1.99		1.52 ± 0.26	1.38–1.66		6.18 ± 0.13	6.11–6.24	

CI: Confidence Level; SD: Standard deviation; *: Two-sample *t*-test.

**Table 3 ijerph-17-02107-t003:** The deviation of the total, longitudinal, and angular errors in four surgical approach by different jaw and dental position.

Tooth Position	Navigation and Surgical Stent		Navigation		Surgical Stent		Freehand		*p*-Value *	Multiple Comparison
	Mean ± SD	95% CI	Mean ± SD	95% CI	Mean ± SD	95% CI	Mean ± SD	95% CI		
Maxillary									<0.0001 #	
Total (mm)	1.12 ± 0.13	1.05–1.19	1.31 ± 0.04	1.29–1.33	1.56 ± 0.05	1.54–1.59	1.82 ± 0.04	1.79–1.84		NS > N > S > F
Longitudinal (mm)	0.66 ± 0.14	0.58–0.74	0.84 ± 0.07	0.80–0.87	1.13 ± 0.08	1.08–1.17	1.21 ± 0.09	1.16–1.25		NS > N > S > F
Angular (degree)	2.43 ± 0.33	2.26–2.61	3.56 ± 0.24	3.43–3.69	4.77 ± 0.07	4.73–4.81	6.02 ± 0.04	6.01–6.05		NS > N > S > F
Mandible									<0.0001 #	
Total (mm)	0.84 ± 0.13	0.78–0.91	1.18 ± 0.07	1.15–1.22	1.44 ± 0.03	1.42–1.45	1.97 ± 0.07	1.93–2.01		NS > N > S > F
Longitudinal (mm)	0.37 ± 0.15	0.29–0.45	0.61 ± 0.06	0.58–0.64	0.89 ± 0.09	0.85–0.94	1.63 ± 0.17	1.53–1.71		NS > N > S > F
Angular (degree)	1.97 ± 0.29	1.82–2.13	2.93 ± 0.12	2.87–2.99	4.31 ± 0.24	4.18–4.44	6.22 ± 0.09	6.17–6.27		NS > N > S > F
Anterior									<0.0001 #	
Total (mm)	0.90 ± 0.18	0.80–1.00	1.20 ± 0.09	1.16–1.25	1.46 ± 0.06	1.43–1.49	1.85 ± 0.07	1.81–1.88		NS > N > S > F
Longitudinal (mm)	0.45 ± 0.21	0.33–0.56	0.68 ± 0.12	0.61–0.74	0.94 ± 0.13	0.87–1.01	1.31 ± 0.19	1.21–1.41		NS > N > S > F
Angular (degree)	2.07 ± 0.37	1.87–2.27	3.10 ± 0.29	2.95–3.26	4.41 ± 0.33	4.24–4.59	6.07 ± 0.09	6.03–6.12		NS > N > S > F
Posterior									<0.0001 #	
Total (mm)	1.06 ± 0.16	0.97–1.14	1.29 ± 0.06	1.26–1.32	1.54 ± 0.08	1.49–1.58	1.94 ± 0.09	1.88–1.99		NS > N > S > F
Longitudinal (mm)	0.59 ± 0.17	0.49–0.68	0.78 ± 0.13	0.71–0.84	1.08 ± 0.13	1.01–1.15	1.52 ± 0.26	1.38–1.66		NS > N > S > F
Angular (degree)	2.34 ± 0.36	2.14–2.53	3.39 ± 0.39	3.17–3.59	4.67 ± 0.18	4.57–4.76	6.18 ± 0.13	6.11–6.24		NS > N > S > F

CI: Confidence Level, *: Using one-way ANOVA; #: Tukey–Kramer HSD; NS: Navigation and Surgical stent; N: Navigation; S: Surgical stent; F: Freehand.

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
