# Peer review of "Comparing Accuracy of Implant Installation with a Navigation System (NS), a Laboratory Guide (LG), NS with LG, and Freehand Drilling"

_ijerph, 2020, doi:10.3390/ijerph17062107_

Round 1
Reviewer 1 Report
The authors compared the accuracy of dental implant surgery with or without the dental navigation system or the surgical guide kit. The experimental design was sound and the conclusion was supported by the results. The reviewer just have one minor comment to the presentation of the results. Since the authors already summarized all the values of the deviation in the tables, they might not need to give the values again in the texts. They could simply compare the different groups in the results by referring to the tables. Also, the units (e.g., mm and degrees) need to be added to the tables.
Reviewer 2 Report
I have some suggestions
In the summary the authors say “A randomized clinical trial was conducted” but in the paper they do not specify how the randomization has been carried out
The hypothesis of this study is that dentists with relatively extensive dental surgery experience do not necessarily have higher accuracy in performing dental implant surgery by using the relevant auxiliary systems. In the discussion the authors say: The current findings confirmed the study hypothesis, but as can be seen in the results and in the conclusions a comparison of the four dental implant surgical methods indicated that the combination of dental implant navigation system and surgical guide kit achieved the highest accuracy
There are duplicate results in the tables and in the text
In the conclusions, I would not include this phrase: “Under the supervision of the Human Research Ethics Committee, each research subject fully understands the purpose and research methods of this research, so as to ensure the rights and obligations of the participants”.
